# Microbiological Evaluation of Water Used in Dental Units

Bartłomiej Błaszczyk [1], Magdalena Pajączkowska [2], Joanna Nowicka [2], Maria Szymonowicz [1], Wojciech Zakrzewski [1,*], Adam Lubojański [1], Marlena Hercuń-Jaskółka [1], Aleksandra Synowiec [1], Sebastian Fedorowicz [2], Wojciech Dobrzyński [3], Zbigniew Rybak [1] and Maciej Dobrzyński [4,*]

[1] Pre-Clinical Research Centre, Wroclaw Medical University, Bujwida 44, 50-345 Wroclaw, Poland; bartlomiej.blaszczyk@student.umw.edu.pl (B.B.); maria.szymonowicz@umw.edu.pl (M.S.); adam.lubojanski@student.umw.edu.pl (A.L.); marlena.hercun-jaskolka@student.umw.edu.pl (M.H.-J.); aleksandra.synowiec@student.umw.edu.pl (A.S.); zbigniew.rybak@umw.edu.pl (Z.R.)

[2] Department of Microbiology, Wroclaw Medical University, ul. Chalubinskiego 4, 50-368 Wroclaw, Poland; magdalena.pajaczkowska@umw.edu.pl (M.P.); joanna.nowicka@umw.edu.pl (J.N.); sebastian.fedorowicz@student.umw.edu.pl (S.F.)

[3] Student Scientific Circle at the Department of Dental Materials, School of Medicine with the Division of Dentistry in Zabrze, Medical University of Silesia in Katowice, Akademicki Square 17, 41-902 Bytom, Poland; wojt.dobrzynski@wp.pl

[4] Department of Pediatric Dentistry and Preclinical Dentistry, Wroclaw Medical University, Krakowska 26, 50-425 Wroclaw, Poland

\* Correspondence: wojciech.zakrzewski@student.umw.edu.pl (W.Z.); maciej.dobrzynski@umw.edu.pl (M.D.); Tel.: +48-51-3182-744 (W.Z.); +48-71-7840-378 (M.D.)

**Abstract:** In modern dentistry, dental units are used for the treatment of patients' teeth, and they need water to operate. Water circulates in a closed vessel system and finally reaches the mucous membranes of the patient as well as the dentist themselves. Therefore, the microbiological safety of this water should be a priority for physicians. This study aims to identify and determine the microbial count, expressed in CFU/mL, in water samples from various parts of the dental unit that are in direct contact with the patient. Thirty-four dental units located in dentistry rooms were analysed. The dentistry rooms were divided into three categories: surgical, conservative, and periodontal. It was found that in surgical rooms, the bacterial count was 1464.76 CFU/mL, and the most common bacterium was *Staphylococcus pasteuri*—23.88% of the total bacteria identified. In dentistry rooms where conservative treatments were applied, the average bacterial concentration was 8208.35 CFU/mL, and the most common bacterium was *Ralsonia pickettii* (26.31%). The periodontal rooms were also dominated by *R. pickettii* (45.13%), and the average bacterial concentration was 8743.08 CFU/mL. Fungi were also detected. *Rhodotorula* spp., *Alternaria* spp., and *Candida parapsilosis* were found to be the most common bacteria which are potentially harmful. This study indicates the need for effective decontamination of the water that is used in dental units and for constant monitoring of the level of contaminants present in the closed vessel system.

**Keywords:** dental unit waterline (DUWL); bacterial contamination; biofilm; water quality

## 1. Introduction

Contamination of the dental unit equipment and water used in it is a significant problem in today's dentistry. This especially poses a risk for immunocompetent patients (e.g., AIDS or cancer patients), patients on chronic corticosteroid treatment, smokers, and for dentists themselves. It should be noted that according to the American Dental Association (ADA), dental water used in dental offices should not contain more than 200 CFU/mL of aerobic, mesophilic, or heterotrophic bacteria.

The origins of dental treatment date back to ancient China. At that time, the treatment involved removal of diseased teeth and not much has changed over the centuries. It was not until the 18th century that the prototype of today's dental chair was created and

improved over the years. Finding the appropriate treatment was supposed to be more beneficial for both the patient and the doctor. The first step was the observation that the patient tolerates dental procedures better when sitting on the chair. The breakthrough came with the invention of a drill powered by the doctor's foot which enabled caries removal. Unfortunately, a low drilling speed and sheer sound of this device caused pain in patients, as well as anxiety and anger. Therefore, in 1953, a team of specialists led by Robert J. Nelson [1] created the first hydraulic, water-cooled, and high-speed drill turbine that alleviated the pain experienced during caries removal, and it additionally reduced the shrill sound. In the following years, the dental unit became more and more similar to the one currently in use, and it was mass-produced by specialised companies rather than by amateur dentists. Today's dental unit consists of a dental chair, water block with a spittoon bowl, high-volume evacuator, saliva ejector, headlight, and control panel along with instruments, such as a turbine dental drill, air and water syringe, and scaler [2], as seen in Figure 1.

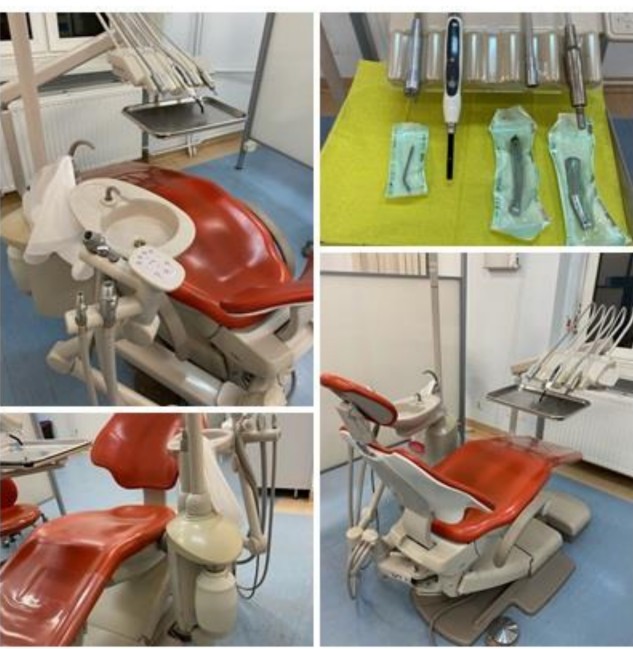

**Figure 1.** Single dental unit with essential accessories.

In order for the dental unit to operate properly, it requires water that usually comes from the water system, and it is used for filling the patient's cup and rinsing the mouth during dental procedures. According to the Polish standards, such water is fit for consumption if it is free from pathogenic microorganisms and parasites in numbers that constitute a potential threat to human health [3]. The water is transported by means of several meters long dental unit waterlines (DUWLs). DUWLs are narrow tubes, approximately 5 mm in diameter, shaped like a drill bit, made of plastics, such as polyvinyl chloride (PVC) or polyurethanes, and less commonly made of glass [4]. Such a design, along with the interruption of the dental unit operation during which water stays in DUWLs, facilitates the growth of microorganisms [5], including *Legionella pneumophila* subsp. *Pneumophila* [6] and *Pseudomonas aeruginosa* [7]. Poorly stored water in an extra container may also contain microorganisms [8]. As early as 1963, there were the first U.K. reports concerning microbial contaminants found in dental units. Studies that focused on the search for the causes of this situation showed the presence of biofilm in DUWLs of dental units [9]. The biofilm, also known as a biological membrane, is a multicellular structure that is composed of various species of viable bacteria, as well as fungi, algae, and protozoa [10]. There are physical and genetic interactions between the microorganisms that make up the biofilm, which ensures a rapid response to changing microbial demands. This structure provides

the organisms living in it with sufficient food, relatively constant external environmental conditions, and protection from antimicrobial compounds. It should be mentioned that the hydrophobic surface of plastic polymers in DUWLs of the dental unit aids the deposition of microorganisms and the formation of biofilm structures [11,12]. The propensity of bacteria to organise into larger structures is so high that even in a newly connected dental unit to the water system, biofilm can form within 8 h and release large numbers of planktonic microorganisms [13].

The second source of microorganisms that can form biofilm is the patient. Microorganisms that are part of the oral microbiota may enter the dental unit component parts along with the patient's saliva, which may be due to the lack of proper valves [14,15].

Widespread testing and quality control of water used in dental procedures is relevant. In Poland, the Chief Sanitary Inspector recommends testing the quality of water from dental equipment at least once a year for the presence of *Escherichia coli*, *P. aeruginosa*, and *Legionella* bacteria, and the total bacterial and microbial—fungal and protozoal—count [16]. Microorganisms found in water can damage equipment and pose a threat to human health. A life-threatening case of pneumonia was reported [17]. The presence of harmful microorganisms in dental units may be proven by the fact that most dentists have elevated levels of antibodies against *Legionella* [18]. There are at least four routes of infection by which microorganisms can cause infection in patients undergoing a dental procedure. These include: hematogenous spread of bacteria, topical contact with the oral mucosa, ingestion of water contaminated with bacteria, and inhalation of bacteria found in the aerosol produced by the operation of dental unit instruments. The bloodborne route is rare in dentistry. Dental procedures may lead to transient oral streptococcal bacteraemia because there is a break in continuity of the oral mucosa, e.g., after tooth extraction or removal. Gastrointestinal disorders caused by ingestion of microorganisms are possible but difficult to link to dental units [19]. The greatest role is played by the patient's inhalation of the aerosol that is produced when dental unit instruments, such as an air syringe, drill or scaler, are cooled [20]. The aerosol contains contaminant particles from DUWLs, which are dispersed in the air. Moreover, the aerosol along with the patient's bodily secretions can persist near the dental chair for up to 24 h after a dental procedure in a single patient [21].

Given all the presented aspects, the problem concerning the contamination of water used in dental units is so serious that it has been considered, and solutions have been sought to reduce the quantity of microorganisms living in these water systems. Particularly intense research began after the outbreak of AIDS caused by the human immunodeficiency virus (HIV) [22], as contact with pathogenic microorganisms during a dental procedure can be harmful for patients with compromised immune systems.

This study aims to perform a microbiological analysis of selected dental unit component parts. The presence of microorganisms was evaluated, their exact location was determined, and, in the case of water samples, a quantitative evaluation was made by determining the number of colony-forming units. The aim of the study is to draw attention to the number of potential pathogens in various places of the dental unit that can cause infection in a patient, especially in those at risk.

## 2. Materials and Methods

### 2.1. Sample Collection

The test material consisted of swabs and water samples taken from dental units located in dentistry rooms at the Academic Dental Polyclinic in Wroclaw. Thirty-four service provided dental units were analysed and classified as surgical (9 units), conservative (17 units), and periodontal (8 units). Swabs were taken from dental unit component parts, such as an air/water syringe, a scaler, a turbine, a micromotor, an inner surface of the bottle under the dental chair, and a spittoon tap (Figure 1). Water samples of 10 mL were taken from a bottle under the dental chair, a cup intended for the patient, a container present in the room, a main bucket of the polyclinic, water used for supplying all dental workstations

excluding surgical ones. Due to the nature of dental procedures, water for dental surgical units is supplied from an external source.

*2.2. Analysis*

The test material was collected using swabs with transport medium. Swabs were cultured on culture media, such as Columbia Agar with 5% sheep blood (Becton Dickinson, Franklin Lakes, NJ, USA), McConkey medium (Biomaxima, Lublin, Poland), and Sabouraud Dextrose with Chloramphenicol LAB-AGAR (Biomaxima, Lublin Poland). The culture for bacteria was conducted for 48 h at 37 °C under aerobic conditions. Fungi were cultured for 10 days at 28 °C. Water samples were taken into sterile Falcon test tubes (Bionovo, Legnica, Poland). Then 1 mL of water was quantitatively cultured (using $10^0$ and $10^{-1}$ dilutions) on culture media, such as Columbia Agar with 5% sheep blood (Becton Dickinson, Franklin Lakes, NJ, USA), McConkey medium (Biomaxima, Lublin, Poland), and Sabouraud Dextrose with Chloramphenicol LAB-AGAR (Biomaxima, Lublin, Poland). After the incubation period was over, the grown colonies of bacteria and fungi were counted manually by authors specialized in microbiology, and the number of colony-forming units per ml of test water (CFU/mL) was determined. The CFU/mL value was calculated according to the following formula:

$$CFU/mL = average\ number\ of\ colonies \times inverse\ of\ dilution \times 10 \tag{1}$$

All cultured microorganisms were identified using MALDI-TOF MS mass spectrometry (Bruker, Billerica, MA, USA).

## 3. Results

The order of dental units in the Table S1 is described by category: surgical, conservative, periodontal.

For the surgical category, *Burkholderia cepacia*, *Staphylococcus warneri* (unit 4, unit 5, unit 6, unit 8), and *Staphylococcus epidermidis* (unit 9) were found to "circulate" within a single unit by passing from bottled water and tap water to instruments of the dentistry panel (air syringe, micromotor). It should be noted that the water supplied to the surgical dental unit room comes from an outside source to ensure that it is sterile.

For conservative dentistry units, where a turbine and scaler are used in addition to an air syringe and micromotor, the presence of *B. cepacia* and *S. epidermidis* was found in both the water used in water systems and in elements used for dental treatment—units 1, 2, and 4.

For the periodontal category, *S. epidermidis* and *Staphylococcus pasteuri* dominated throughout unit 4. The mould fungi were found in 27 out of 34 dental units analysed. First and foremost, the fungi were present in a bottle under the dental chair, where the water that supplies the entire system is taken from. In the surgical units, *Cladosporium* spp. was found in a bottle that supplies the dental unit with water in six out of nine dental units. High concentrations of mould fungi were found in some of the units that belong to the conservative category (Figure 2). The bottle with water of each of these dental units was dominated by *Alternaria* spp., *Cladosporium* spp., and *Fusarium oxysporum*.

For further analysis, the aggregation of results was obtained from the quantitative assessment of sampled water used for the operation of all dental units in surgical rooms. The total bacterial contamination of water was 73,238 CFU/mL. The lowest, average, and highest concentrations of bacteria at individual sampling sites of the dental unit were 10, 1464.76 and 13,500 CFU/mL, respectively. Twenty-seven bacterial species were isolated, of which *S. pasteuri* was the most common bacterium (23.88% of all bacteria) as well as *S. warneri*—20.74% and *Cupriavidus metallidurans*—12.83%. In terms of prevalence, the next microorganisms were *Micrococcus luteus*—9.87%, *Ralstonia pickettii*—6.62%, *Leifsonia shinshuensis*—6.28%, *Clostridium novyi*—5.19%, *Staphylococcus cohnii*—4.06%, *Lactobacillus paracasei*—2.75%, and *Bacillus mycoides*—1.50%. Other microorganisms were <1% of the total bacteria: *S. epidermidis*—0.75%, *Streptococcus parasanguinis*—0.68%, *B. cepacia*—0.61%,

*Kocuria palustris*—0.56%, *Cutibacterium avidum*—0.55%, *Escherichia coli*—0.44%, *Streptococcus oralis*—0.41%, *Pseudarthrobacter oxydans*—0.33%, *Microbacterium esteraromaticum*—0.30%, *Stenotrophomonas maltophilia*—0.27%, *Actinomyces dentalis*—0.27%, *Aromatoleum toluolicum*—0.23%, *Brevundimonas aurantiaca*—0.22%, *Microbacterium testaceum*—0.22%, *Brevundimonas diminuta*—0.19%, *Microbacterium ginsengisoli*—0.14%, and *Cupriavidus pauculus*—0.11%. The pathogenicity of the mentioned microorganisms is presented in Table 1.

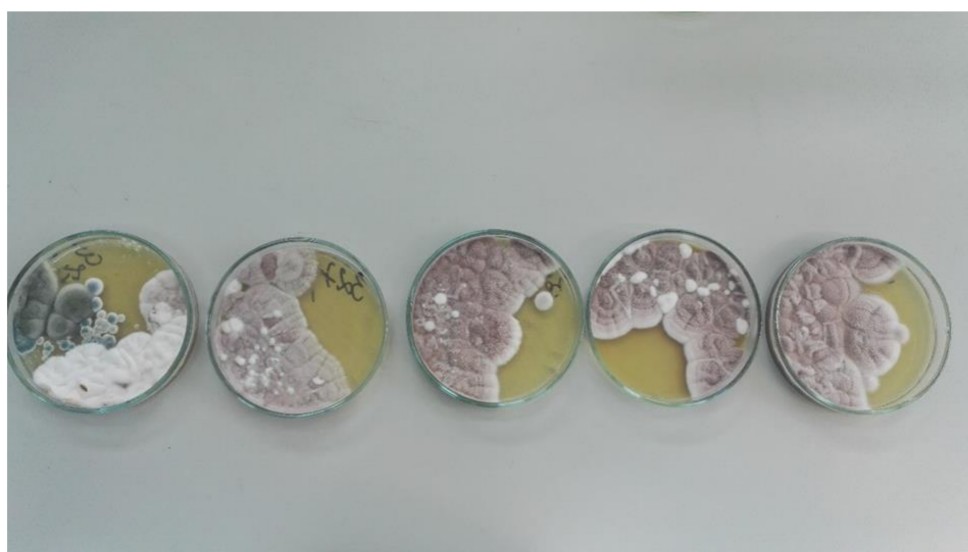

**Figure 2.** Mould fungi isolated from 5 dental units used for conservative dentistry.

**Table 1.** Microorganisms isolated from water samples and their potential pathogenicity.

| Microorganisms | Presence in Dental Units | Proportion [%] | Pathogenicity |
|---|---|---|---|
| Gram-positive bacteria | | | |
| *A. dentalis* | 1 | 2.9 | periodontitis [23] |
| *A. oris* | 1 | 2.9 | oral microbiota [23] |
| *B. casei* | 2 | 5.9 | opportunistic pathogen that causes peritonitis in immunocompromised patients [24] |
| *B. celere* | 1 | 2.9 | no literature data available |
| *C. avidum* | 1 | 2.9 | opportunistic pathogen that is present primarily in immunocompromised patients receiving chemotherapy and radiotherapy (cutaneous abscesses, infective endocarditis), deep tissue infections [25]; potential aetiological agent of periprosthetic joint infection [26] |
| *K. palustris* | 3 | 8.8 | bacteraemia, endocarditis, peritonitis, cholecystitis, urinary tract infections, brain abscesses, and keratitis [27] |
| *K. kristinae* | 1 | 2.9 | bacteraemia associated with the use of central venous catheter, infective endocarditis, acute peritonitis, abdominal abscesses, umbilical sepsis, acute cholecystitis, and urinary tract infection [28] |
| *L. shinshuensis* | 13 | 38.2 | Combined with other bacterial species, they cause infections associated with the use of central venous catheters used as vascular access to haemodialysis [29]. |
| *M. phyllosphaerae* | 2 | 5.9 | plant pathogen; a case of prosthetic hip infection has been described [30] |

**Table 1.** *Cont.*

| Microorganisms | Presence in Dental Units | Proportion [%] | Pathogenicity |
|---|---|---|---|
| *M. ginsengisoli* | 5 | 14.7 | no literature data available |
| *M. testaceum* | 3 | 8.8 | blood poisonings, urinary tract infections [30] |
| *M. esteraromaticum* | 2 | 5.9 | blood poisonings [30] |
| *R. dentocariosa* | 1 | 2.9 | caries, infections of paradental tissues and opportunistic infections: endocarditis, peritonitis, sepsis, lung abscesses, pneumonia in patients with immune deficiencies, neoplasms [31] |
| *R. amarae* | 4 | 11.76 | bloodstream infections in haematological patients with comorbidities [31] |
| Gram-negative bacteria | | | |
| *A. ursingii* | 3 | 8.8 | bacteraemia in patients with haematological neoplasms and neutropenia [32] |
| *A. lwoffii* | 1 | 2.9 | microbiota of the oropharynx, skin, and perineum in approx. 25% of healthy individuals; opportunistic pathogen in immunocompromised patients and nosocomial infections: sepsis, pneumonia, meningitis, and wound infections [33,34] |
| *A. citratiphilum* | 1 | 2.9 | occurs in water systems [35]; no reports concerning pathogenicity |
| *B. fragilis* | 1 | 2.9 | peritonitis; soft tissue infections; pelvic, lung, and brain abscesses [36] |
| *B. cepacia* | 9 | 26.47 | opportunistic pathogen in patients with cystic fibrosis [37], involved in nosocomial infections |
| *B. pyrrocinia* | 1 | 2.9 | no literature data available |
| *C. metallidurans* | 8 | 25.53 | catheter-related infections [38] |
| *C. pauculus* | 2 | 5.9 | bacteraemia in haematological, oncologic, and diabetic patients and those with chronic heart failure [38] |
| *E. coli* | 1 | 2.9 | pathogenic serotypes may cause bloody diarrhoea, hemolytic uremic syndrome, acute renal failure, and sepsis [39] |
| *M. morganii* | 1 | 2.9 | infections in immunocompromised patients; nosocomial infections [40] |
| *N. resinovorum* | 1 | 2.9 | They are found in lake, soil, and water [41]. |
| *P. septica* | 1 | 2.9 | bacteraemia in immunocompromised patients, those with comorbidities and with vascular catheters [42] |
| *P. fungorum* | 1 | 2.9 | opportunistic bacterium [43] |
| *P. putida* | 2 | 5.9 | opportunistic pathogen causing nosocomial infections mainly in neutropenic patients, cancer patients, neonatal patients, and those with cystic fibrosis [44] |
| *P. xantuomarina* | 1 | 2.9 | no literature data available |
| *R. pickettii* | 7 | 20.5 | Waterborne opportunistic pathogen [45]. Transmission may occur due to contaminated medical devices or solutions typical of health care facilities (post-dialysis sepsis) [46,47]. The source may be a contaminated, intravascular solution intended for haematological patients [45]. This causes diseases in patients with compromised immunity, respiratory infections—especially in patients with cystic fibrosis, meningitis, peritonitis, and sepsis. |

**Table 1.** *Cont.*

| Microorganisms | Presence in Dental Units | Proportion [%] | Pathogenicity |
|---|---|---|---|
| *S. maltophilia* | 7 | 20.5 | It mainly causes nosocomial infections (bacteraemia, biliary tract infections, urinary tract infections, respiratory infections, skin and soft tissue infections, endocarditis) [48,49]. Nosocomial infections, usually in those with comorbidities (e.g., malignancy, HIV) and with chronic lung diseases or lung cancer. This species was classified as an ESKAPE pathogen due to its high level of drug resistance [48]. |
| *A. russicus* | 1 | 2.9 | no literature data available |
| Gram-positive granulomas | | | |
| *E. faecalis* | 1 | 2.9 | dental root canal infections [50]; aetiologic agent of endocarditis; bacteraemia in patients with comorbidities [51] |
| *M. luteus* | 8 | 25.53 | opportunistic pathogen that causes infections such as endocarditis or bacteraemia in cancer patients [52] |
| *P. polychromogenes* | 2 | 5.9 | no literature data available |
| *P. oxydans* | 1 | 2.9 | no literature data available |
| *S. cohnii* | 3 | 8.8 | catherer-related infections, meningitis, urinary tract infections, and cholecystitis [53,54]; valve endocarditis in immunocompetent patients [55] |
| *S. haemolyticus* | 10 | 29.41 | opportunistic nosocomial pathogen—bacteraemia, meningitis, skin infections, prosthetic joint infections, and endocarditis [56] |
| *S. capitis* | 7 | 20.5 | opportunistic nosocomial pathogen that causes bloodstream infections, joint and valve prosthesis infections, bacterial endocarditis with subsequent osteomyelitis [57] |
| *S. epidermidis* | 23 | 67.65 | skin microbiota; an important opportunistic pathogen that causes bloodstream infections, endocarditis, osteomyelitis, and abscesses in immunocompromised patients; these infections are associated with the use of intravascular catheters, valves, prosthetic heart valves, artificial lenses, and orthopaedic implants [58]. |
| *S. hominis* | 5 | 14.7 | potentially opportunistic and nosocomial pathogen that causes infections in patients with compromised immune systems (infective endocarditis) [59] |
| *S. pasteuri* | 5 | 14.7 | opportunistic pathogen in immunocompromised patients—bacteraemia in patients with leukemia [60]; urinary tract infection associated with the use of a urinary catheter in a patient receiving chemotherapy; endocarditis [61]; osteomyelitis [62] |
| *S. warneri* | 20 | 58.82 | infections in immunocompromised patients (endocarditis, sepsis, septic arthritis, meningitis, cutaneous botryomycosis) [63]; however, there are cases of endocarditis in an immunocompetent patient (early and late endocarditis) after tooth extraction or mammoplasty [64]. |
| *S. aureus* | 5 | 14.7 | causes pneumonia, endocarditis, osteomyelitis, and sepsis in hospitalised and immunocompromised patients [58] |
| *S. xylosus* | 2 | 5.9 | Late knee joint infection in a patient who underwent total knee alloplasty 18 years earlier; commensal bacterium associated with skin and mucous membranes, rarely involved in infections [65] |
| *S. succinus* | 1 | 2.9 | no literature data available |

**Table 1.** *Cont.*

| Microorganisms | Presence in Dental Units | Proportion [%] | Pathogenicity |
|---|---|---|---|
| *S. parasanquinis* | 1 | 2.9 | It causes endocarditis. It causes infections in 10% of oncological patients; peritonitis was found [66]. |
| *S. sanguinis* | 2 | 5.9 | subacute infective endocarditis; septic arthritis; in patients with periodontal disease, endocarditis, or valvular heart disease [58] |
| *S. oralis* | 2 | 5.9 | low pathogenicity; in rare cases it may cause meningitis in patients after dental procedures and with poor oral hygiene [24]; rare cases of peritonitis associated with peritoneal dialysis [67] |
| *S. salivarius* | 1 | 2.9 | exogenous and endogenous intraocular inflammation in patients with comorbidities [68] |
| Gram-positive bacilli (red, spore-forming) | | | |
| *B. pumilus* | 6 | 17.6 | septic arthritis in healthy individuals [69] and skin infections [70]; sepsis in a newborn was described in the literature [71]. |
| *B. mycoides* | 3 | 8.8 | blood poisoning described in the literature [72] |
| *B. weihenstephanensis* | 3 | 8.8 | non-pathogenic, dairy contamination [73] |
| *B. cereus* | 3 | 8.8 | Bacteraemia in immunocompromised patients and haematological malignancies [71] |
| *B. simplex* | 4 | 11.76 | no literature data available |
| *B. muralis* | 3 | 8.8 | no literature data available |
| *B. megaterium* | 2 | 5.9 | considered non-pathogenic or at least of very low malignity [74]; however, it may cause pleurisy, inflammation of the eye, dermatitis, and brain abscesses. |
| *C. novyi* | 1 | 2.9 | aetiological agent of severe soft tissue infections in injection heroin users; bacteraemia and hepatic gas gangrene in a patient with gastric cancer and diabetes [75,76] |
| *L. boronitolerans* | 1 | 2.9 | no literature data available |
| *L. coleohominis* | 1 | 2.9 | considered non-pathogenic; isolated from urine and cervix without clinical description [77] |
| *L. paracasei* | 1 | 2.9 | caries, sepsis, pneumonia, infective endocarditis, or splenic abscesses in immunocompromised patients [78] |
| *Paenibacillus* spp. *(Gram-positive* or *Gram-variable)* | 1 | 2.9 | no literature data available |
| Gram-negative bacilli | | | |
| *B. aurantiaca* | 2 | 5.9 | Usually nosocomial infections. The role of this microorganism in human pathogenicity needs further studies but it was linked to two cases of bloodstream infection [79]. |
| *B. diminuta* | 3 | 8.8 | nosocomial infections; peritonitis in a patient with end-stage renal failure; lower leg ulceration in a patient with glomerulonephritis; keratitis; pleurisy; bacteraemia in diabetic and haematological patients [79] |
| *Brevundimonas* spp. | 1 | 2.9 | nosocomial infections [79]. |

| Microorganisms | Presence in Dental Units | Proportion [%] | Pathogenicity |
|---|---|---|---|
| *S. paucimobilis* | 1 | 2.9 | nosocomial (hospital-acquired) and community-acquired infections in patients with chronic diseases and immunosuppressed patients; sepsis, septic pulmonary embolism, septic arthritis, and intraocular inflammation [14,80], wound infections [8,20], septic shock in a patient with burns [4], catheter-related bacteraemia [19], pneumonia [19], splenic abscesses [31], urinary tract infection, empyema [10], peritonitis [5,8,12,19,30]. There were also cases of invasive infections such as meningitis, osteomyelitis [33]. It was reported that *S. paucimobilis* was isolated from maxillary sinus washouts in four patients, as the irrigation solution (saline) was contaminated with this microorganism [81]. |
| Fungi and mould | | | |
| *C. parapsilosis* | 2 | 5.9 | opportunistic pathogen, superficial and systemic candidiasis in oncological patients [82] |
| *Rhodotorula* spp. | 1 | 2.9 | fungemia in patients with impaired immune system function; catheter-related sepsis [82]; endocarditis, meningitis [83] |
| *Alternaria* spp. | 17 | 50 | sinusitis, skin infections in immunosuppressed patients [82] |
| *A. niger* | 3 | 8.8 | ear infection, pulmonary aspergillosis in immunocompromised patients [81,82], sinusitis [84] |
| *Cladosporium* spp. | 9 | 26.47 | rarely pathogenic [80] |
| *Fusarium* spp. | 2 | 5.9 | sinusitis in patients with leukemia [82,84], keratitis, pneumonia, hematogenous spread [84] |
| *F. oxysporum* | 13 | 38.2 | isolated from the blood of a patient with leukemia and invasive fusariosis [82] |
| *Penicilum* spp. | 4 | 11.76 | keratitis, pneumonia, endocarditis [85] |
| Other | | | |
| *A. Toluolicum* | 1 | 2.9 | no literature data available |

The total fungal contamination of water in dental surgical units was 20,400 CFU/mL. The lowest concentration at individual sampling sites of the dental unit was 10 CFU/mL, while the highest one was 19,900 CFU/mL. Five types of fungi were cultured. The most commonly cultured fungus was *Rhodotorula* spp.—97.56%. Other isolated fungi included *Cladosporium* spp.—1.37%, *Alternaria* spp.—0.49%, *Penicium* spp.—0.49%, and *Fusarium oxysporum*—0.09%.

For conservative dentistry units, the total bacterial contamination of water was 44,3251 CFU/mL. The lowest single concentration of bacteria in a given part of the dental unit was 90 CFU/mL, the average was 8208.35 CFU/mL, and the highest one was 64,600 CFU/mL. Twenty-one bacterial species were cultured, with the highest water contamination caused by *R. pickettii* (26.31% of all bacteria). Bacteria that are also common include *Bacillus pumilus*—18.04%, *Aquabacterium citratiphilum*—14.57%, and *C. pauculus*—11.73%. Other bacteria were present in much lower quantities: *S. epidermidis*—9.52%, *M. luteus*—9.10%, *S. pasteuri*—2.52%, *S. capitis*—2.08%, *R. amarae*—1.88%, and *B. diminuta*—1.06%. Microorganisms that represent <1% of the total bacteria include *B. cepacia*—0.74%, *C. metallidurans*—0.72%, *S. haemolyticus*—0.42%, *S. warneri*—0.41%, *S. aureus*—0.40%, *Bacillus simplex*—0.27%, *Acinetobacter lwoffii*—0.07%, *L. shinshuensis*—0.06%, *S. maltophilia*—0.06%, *Kocuria kristinae*—0.02%, and *Lysinibacillus boronitolerans*—0.02%.

The total fungal contamination of water in this category was 140 CFU/mL. The lowest concentration at individual sampling sites of the dental unit was 40 CFU/mL, while the

highest one was 100 CFU/mL. One species of fungus, *Candida parapsilosis*, was cultured (100%).

In the periodontal category, the total bacterial contamination of water was 11,3660 CFU/mL. The lowest single concentration of bacteria in a given part of the dental unit was 50 CFU/mL, the highest one was 43,000 CFU/mL, and the average was 8,743.08 CFU/mL. Six bacterial species were found in the samples. *R. pickettii*—45.13%, *S. epidermidis*—39.23%, and *S. pasteuri*—14.43% were the predominant bacteria identified. Other bacteria were found in much smaller quantities: *Bacillus* simplex—0.93%, *Pantoea septica*—0.19%, and *S. succinus*—0.09%. The total fungal contamination of water in this category was 350 CFU/mL. The lowest concentration at individual sampling sites of the dental unit was 10 CFU/mL, while the highest one was 340 CFU/mL. The cultured fungi included *Alternaria* spp.—97.14% and *Penicilium* spp.—2.86%.

The analysis of the quantity of individual microorganisms colonising dental units revealed that in most cases their quantity exceeded the permissible standard. A list of dental units where the quantity of bacteria in 1ml of water exceeded the limit value of CFU/mL is shown in Table 2.

**Table 2.** Microorganisms exceeding the limit value of CFU/mL in dental units.

| Category | Unit | Bottle under the Dental Chair | Cup CFU/mL | Water Container in the Room |
|---|---|---|---|---|
| Surgical | 1 | | *S. cohnii* $2.7 \times 10^3$ *C. novyi* $3.8 \times 10^3$ | *S. warneri* $8.5 \times 10^3$ |
| | 1 | | | |
| | 2 | *M. luteus* $7 \times 10^3$ *S. pasteuri* $1.35 \times 10^4$ | | |
| | 4 | | *S. pasteuri* $3.4 \times 10^3$ *S. warneri* $2.7 \times 10^3$ | |
| | 6 | *Rhodotorula* spp. $1.99 \times 10^4$ | | |
| | 7 | *R. pickettii* $1.4 \times 10^3$ | | |
| | 8 | | *B. mycoides* $1.1 \times 10^3$ *L. shinshuensis* $4.2 \times 10^3$ *C. metallidurans* $6.4 \times 10^3$ | |
| Conservative | 1 | *C. metallidurans* $1.3 \times 10^3$ *B. cepacia* $6 \times 10^2$ | | *R. pickettii* $4.3 \times 10^4$ *S. epidermidis* $1.2 \times 10^3$ |
| | 1 | | | |
| | 2 | *B. cepacia* $1.3 \times 10^3$ | *B. simplex* $1.2 \times 10^3$ *S. aureus* $4.6 \times 10^2$ | |
| | 2 | | *S. epidermidis* $7.1 \times 10^3$ | |
| | 1 | | | |
| | 2 | | *S. epidermidis* $1.02 \times 10^4$ *B. pumilus* $2.08 \times 10^4$ | |
| | 3 | | *M. luteus* $2.6 \times 10^4$ | |
| | 4 | *R. picketti* $1.0 \times 10^3$ | *S. epidermidis* $1.14 \times 10^4$ | |
| | 3 | | *S. epidermidis* $6.1 \times 10^3$ | |
| | 1 | | | |
| | 2 | *M. luteus* $2.2 \times 10^3$ | *B. pumilus* $3.00 \times 10^4$ | |
| | 4 | *R. picketti* $3.3 \times 10^3$ *S. warneri* $1.6 \times 10^3$ *S. haemolyticus* $1.6 \times 10^3$ *C. metallidurans* $1.9 \times 10^3$ | *R. amarae* $7.6 \times 10^3$ | |
| | 5 | *M. luteus* $1.2 \times 10^3$ *B. cepacia* $1.4 \times 10^3$ | | |
| | 4 | *B. pumilus* $2.13 \times 10^4$ *M. luteus* $8.8 \times 10^3$ | | *R. picketti* $4.7 \times 10^3$ *S. aureus* $1.09 \times 10^3$ |
| | 1 | | | |
| | 2 | *C. pauculus* $3.8 \times 10^4$ | *M. luteus* $2.1 \times 10^3$ *C. pauculus* $1.4 \times 10^4$ | |
| | 3 | | *S. epidermidis* $2.2 \times 10^3$ *S. capitis* $9.2 \times 10^3$ *B. diminuta* $4.7 \times 10^3$ | |
| | 4 | *R. pickettii* $6.46 \times 10^4$ *R. amarae* $5.01 \times 10^2$ *S. epidermidis* $3.5 \times 10^3$ | *S. pasteuri* $1.12 \times 10^4$ *A. citratiphilum* $6.46 \times 10^4$ *B. pumilus* $7.6 \times 10^3$ | |

**Table 2.** *Cont.*

| Category | Unit | Bottle under the Dental Chair | Cup CFU/mL | Water Container in the Room |
|---|---|---|---|---|
| Periodontal | 1 | | *B. simplex* $6 \times 10^2$ | *S. epidermidis* $1.2 \times 10^4$ |
| | 2 | | | *R. pickettii* $2.9 \times 10^3$ |
| | 2 | | *S. epidermidis* $2.9 \times 10^4$ | *R. pickettii* $5.4 \times 10^3$ |
| | 4 | | *S. pasteuri* $1.64 \times 10^4$ | *S. epidermidis* $2.34 \times 10^3$ |

The proportion of each microorganism against all DUWLs and its potential pathogenicity are shown in Table 2. *S. epidermidis* (67.65), *S. warneri* (58.82), and *Alternaria* spp. were the most frequently isolated microorganisms in 34 dental units (50). *Fusarium oxysporum* and *Leifsonia shinshuensis* were equally prevalent (38.2), as were *Cladosporium* spp. and *Burholderia cepacia* (26.47). *Staphylococcus haemolyticus* (29.41) and *M. luteus* (25.53) showed a proportion of more than 25% of all dental units.

## 4. Discussion

### 4.1. International Dental Water Guidelines and Our Research Result

As this study reveals, dental unit water and equipment contamination is a serious problem in modern dentistry. In contrast, according to common ADA and Centers for Disease Control and Prevention (CDC) guidelines, the maximum contamination of water used for dental treatment should be less than 500 CFU/mL [86]. In terms of analysed surgical, periodontal, and conservative dentistry units at the Academic Dental Polyclinic in Wroclaw, microbial contamination far exceeded the threshold of 500 CFU/mL for most of the water samples. Only 46% of water sampling sites of the dental unit did not exceed the above-mentioned bacterial count limit. Driven by ADA guidelines alone, only 29% of dental workstations met the requirements. The quantity of bacteria such as *S. warneri*, *S. pasteuri*, *R. picketti*, *S. epidermidis,* and *B. pumilus* exceeded 10,000 CFU/mL, which is the majority of microorganisms found in the water samples tested. These bacteria are potentially pathogenic to humans, especially at such high concentrations. They may cause osteoarthritis, dermatitis, inflammation of the eyes/ears in healthy patients, as well as bacteraemia.

### 4.2. Comparison to the Results of Research by Other Authors

Uzel et al. [87] revealed that in terms of microbial contaminants of DUWLs, *B. 13epacian*, *Chryseomonas luteola*, *Pseudomonas fluorescens*, *R. pickettii*, and *Sphingomonas paucimobilis* were the most abundant among the isolated and identified microorganisms. Similar results were obtained in the present study. In terms of the prevalence of bacteria in 34 dental units, by A-dec, Newberg, Oregon, U.S.A., *S. epidermidis*, *S. warneri*, *Alternaria* spp., *F. oxysporum*, *L. shinshuensis*, *Cladosporium* spp., and *B. cepacia* were most frequently isolated. According to Szymańska et al. [88], the most common contaminants in Poland include *Ralstonia pickettii*, *Staphylococcus* spp., *Sphingomonas paucimobilis*, *Actinomyces* spp., and *Micrococcus* spp. In this study, the bacteria such as *Staphylococcus* spp., *R. pickettii*, and *Micrococcus* spp. also appeared in very large numbers, while *Sphingomonas paucimobilis* and *Actinomyces* spp. were found in single dental units. Interestingly, no other studies have reported the presence of *L. shinshuensis* in DUWLs. Reports by Arvand et al. [89] and Gawish et al. [90] found the presence of the previously mentioned, potentially dangerous bacteria, such as *Legionella pneumophila* subsp. *Pneumophila* and *P. aeruginosa*. In this study, the bacteria in question were not present, although there were bacilli of the genus *Pseudomonas*—*P. septica*, *P. fungorum*, *P. putida*, and *P. xantuomarina*. According to the category of room where the dental units are located, the average bacterial concentrations are (1464.76, 8208.35, 8743.08) CFU/mL, respectively. The presented contamination values are many times higher than the acceptable microbial standards according to the ADA and CDC, while studies by Szymańska et al. [88] and Souza-Gugelminet al. [91] obtained much higher bacterial concentrations in most cases compared to the results of this study. The present study revealed the presence of fungi and mould, which were found in 27 out of 34 dental units

representing 79% of all units analysed. *Candida parapsilosis* was present in 2 dental units, *Rhodotorula* spp. in 1 dental unit, *Clostridium* spp. in 10 dental units, *Aspergillus niger* in 2 dental units, *Alternaria* spp. in 17 dental units, *Penicilium* spp. in 4 dental units, whereas *Fusarium oxysporum* was found in 12 dental units. Similar results, i.e., the presence of fungi in 10 out of 18 analysed dental units (55.56%), were obtained by Mazarii et al. [92]. In studies by Lisboa et al. [93], *Aspergillus*, *Penicilum*, *Cryptococcus*, and *Candida guilliermondii* were detected in 7 dental units out of 41 ones evaluated. Damasceno et al. identified fungi, such as *Aspergillus* spp., *Fusarium* spp., *Candida* spp., and *Rhodotorula* spp. The number of fungal colonies at different points of the dental unit ranged from 0 to 40 CFU/mL [94]. In this study, the amounts of fungi range from 10 to 19,900 CFU/mL.

*4.3. Methods for Improving the Quality of Dental Water*

The water supplying the dental surgical units is sterile water that is supplied to the facility from an outside source. Pankrust et al. [11] recommends the use of sterile water in each surgical procedure. This study reveals that in the dental surgical units supplied by deionised water, the microbial species diversity is significantly higher compared to the periodontal or conservative dentistry units that are supplied by plain distilled water. Studies by Walker [14] and Rickard [95] found no significant differences between DUWL systems filled with distilled water from a reservoir bottle or deionised water in surgical rooms [14,95].

The unit water supply bottles are usually manually filled with water (tap water, distilled water, or sterile water), whereby the contamination of water was from skin bacteria, such as *S. epidermidis* and *S. aureus.* This may explain why *S. epidermidis* was most frequently isolated in this study. Bottles should be cleaned and disinfected regularly to eliminate such situations. Preferably, reservoir bottles should be sterilised regularly in an autoclave before being refilled and reused [96].

However, there are no standards or regulations that specifically address the microbiological quality of water used in dental units. This is because DUWLs are considered medical devices in which water is intended for the proper operation of these devices, such as cooling and irrigation of instruments from a dentistry panel. During use, water is ingested by patients in small amounts. Moreover, the aerosol produced by dental instruments is inhaled [96]. Systematic disinfection of these dental devices is necessary. Chemicals based on hydrogen peroxide, silver ion hydrogen peroxide, chlorhexidine gluconate, sodium hypochlorite, peracetic acid, citric acid [96], glutaraldehyde, chlorhexidine, or chlorine dioxide are most commonly used for reducing contamination in the dental unit. Physical methods such as filtration or reverse osmosis can also be used [97]. The periodic or intermittent use of chemicals in question effectively reduced microbial counts in water samples below 200 CFU/mL. Therefore, they appear to be currently the best choice for unit decontamination. Unfortunately, these chemicals may chemically react with component parts of the dental unit, i.e., various valves or DUWL elements, which cause their damage or corrosion.

Less effective ways to prevent high microbial counts include providing sterile water and using it to operate the dental unit and flushing DUWLs regularly with water. The reason for the lack of effectiveness of these methods is that even sterile water flowing through the biofilm that is formed in DUWLs transfers microorganisms living in it. The provision of dental unit instruments with valves that prevent the patient's oral bacterial flora from reverse entering into DUWLs are not sufficient to maintain sterility, as these valves often fail [96]. Researchers who investigate the problem of water contamination in the dental unit point to the need to improve both the patients' and dentists' awareness of possible infections due to microbial contamination in dental offices. They also emphasise that it is essential to develop and use effective methods for eradication of microorganisms living in DUWLs [88], as well as training in terms of the proper use and maintenance of these systems. There are currently no established regulations for inspection and disinfection of dental units. The manufacturers of this equipment, as well as the users, dentists, and

support staff, do not have adequate knowledge of water quality and the role of biofilm in these water systems, while the presence of significant numbers of microorganisms is an alarming problem.

## 5. Conclusions

The specific structure of dental units contributes to form biofilm and microbial contamination of the dental unit waterlines (DUWLs). The presence of bacteria in water supply tubes, including DUWLs, is a common phenomenon, which has been well documented around the world. Our study shows that the amounts of the most frequently found microorganisms were above the limits. This is a potential risk for many immunosuppressed patients and doctors, which may lead to dangerous infections during normal dental treatments. For this reason, researchers and physicians need to control and prevent the bacterial biofilms in DUWLs by using appropriate and effective methods of disinfection of the entire dental unit chairs.

**Supplementary Materials:** The following supporting information can be downloaded at: https://www.mdpi.com/article/10.3390/w14060915/s1, Table S1: Microorganisms isolated from individual dental unit component parts and the water supplying these systems, and CFU/mL.

**Author Contributions:** Z.R. and M.D. conceived the design of the experiments and analyzed the data; M.S. contributed reagents/materials/analysis tools and analyzed the data; M.S., M.D. and Z.R. participated in funding acquisition; J.N. and M.P. designed and performed the microbiological experiments and analyzed the data; B.B. analyzed the data; B.B., W.Z., A.L., M.H.-J., A.S., S.F. and W.D. performed the microbiological experiments. All authors contributed to the writing of the paper. All authors have read and agreed to the published version of the manuscript.

**Funding:** The funding was supported by Vice-Rector for Educational Affairs of Wrocław Medical University.

**Institutional Review Board Statement:** Not applicable.

**Informed Consent Statement:** Not applicable.

**Data Availability Statement:** Not applicable.

**Acknowledgments:** Acknowledgments This study was performed as part of the Student Scientific Circle of the Experimental Dentistry and Biomaterials Research (K145) and the Microbiology Students Association (K34).

**Conflicts of Interest:** The authors declare that there are no conflicts of interest regarding the publication of this paper.

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
