# Peer review of "Microbiological Evaluation of Water Used in Dental Units"

_water, doi:10.3390/w14060915_

Round 1
Reviewer 1 Report
The authors have investigated the presence of micro-organisms in different component parts of a dental unit. Their special attention was water that dentists use during the dental treatments.
The introduction section is well written with enough background information which leads it to the goals of this study. However, I felt that introduction was ended abruptly. I would suggest you add a line or two after line 123 to mention very briefly the reasons of this study. You have mentioned the goals (what are you doing in this study) but didn’t mention what will come out of the findings.
Line 145: counted by what?
Section: Materials and methods: You could divide this section in to two parts a) Sample collection b) analysis. It should read better doing that. Otherwise, this section has enough details.
The results section is also well written with proper presentation of data through Tables.
Line 256-257: Revise this sentence.
Discussion section can be divided in least two subsections for better reading. There should a subsection for the recommendations based on your findings.
Author Response
Dear Reviewer,
We would like to express our sincerest gratitude to the Reviewers for their enormous efforts in criticizing the manuscript. All remarks have been included in the revised version of the manuscript.
Question 1
The introduction section is well written with enough background information which leads it to the goals of this study. However, I felt that introduction was ended abruptly. I would suggest you add a line or two after line 123 to mention very briefly the reasons of this study. You have mentioned the goals (what are you doing in this study) but didn’t mention what will come out of the findings.
Answer: We would like to thank you for the comment. The introduction part has been extended according to Reviewers suggestions.
Question 2
Line 145: counted by what?
Answer: We would like to thank you for the comment. Colonies of bacteria and fungi have been counted by authors specialized in microbiology. This information has been added to the manuscript.
Question 3
Section: Materials and methods: You could divide this section in to two parts a) Sample collection b) analysis. It should read better doing that. Otherwise, this section has enough details.
Answer: We would like to thank you for the comment, Sections have been added to the manuscript.
Question 4
Line 256-257: Revise this sentence.
Answer: We would like to thank you for the comment, The sentence has been revised.
Question 5
Discussion section can be divided in least two subsections for better reading. There should a subsection for the recommendations based on your findings.
Answer: We would like to thank you for the comment. The discussion part has been divided.

Reviewer 2 Report
Abstract
Line no-35: There are many types of bacteria are present in water. Please, mention whether these all species of bacteria are harmful or not?
Introduction
Line no-98: Please replace the word 'hazardous" with 'harmful'.
Materials and Methods
Line no-127: Please add 'service provided' dental units.
Result
Line no-172: Are these bacteria harmful? Mention the disease caused by these bacteria?
Line no-180: Please summarize the major pathogen only, and make a table of these bacteria isolated from samples of major units at the annex.
Line no-237: Please mention the summary information of this table 3 in the result section and keep it as it is in the annex section.
Discussion
Line no-247: Better to move these first 3 sentences into the introduction section.
Line no-255: Please delete these words "pathogens that are".
Line no-274: Please mention the name of the dental units in the sentence.
Reference
Line no-372, 442, and 491: Please re-check your references 3, 31, and 49 and write properly.
Line no-544: Year and DOI are missing. Please write it properly.
Author Response
Dear Reviewer,
We would like to express our sincerest gratitude to the Reviewers for their enormous efforts in criticizing the manuscript. All remarks have been included in the revised version of the manuscript.
Reviewer #2
Abstract
Question 1
Line no-35: There are many types of bacteria are present in water. Please, mention whether these all species of bacteria are harmful or not?
Answer: We would like to thank you for the comment. The information about harmfulness of bacteria has been added to the manuscript.
Introduction
Question 2
Line no-98: Please replace the word 'hazardous" with 'harmful'.
Answer: We would like to thank you for the comment. The word has been replaced.
Materials and Methods
Question 3
Line no-127: Please add 'service provided' dental units.
Answer: We would like to thank you for the comment. It has been added to the manuscript.
Result
Question 4
Line no-172: Are these bacteria harmful? Mention the disease caused by these bacteria?
Answer: We would like to thank you for the comment. Additional information has been added to the manuscript.
Question 5
Line no-180: Please summarize the major pathogen only, and make a table of these bacteria isolated from samples of major units at the annex.
Answer: When discussing the results, the authors mention only the most frequently isolated species. 10 strains appear with a frequency of 24-1.5%, and 17 other strains occur with a frequency below 1%. The pathogenicity of all isolated microorganisms is presented in Table 3. According to the authors of the study, it is important to present all types of microorganisms grown from the collected samples. The pathogenicity of the mentioned microorganisms is presented in Table 3.
Question 6
Line no-237: Please mention the summary information of this table 3 in the result section and keep it as it is in the annex section.
Answer:We would like to thank you for the comment. The authors agree that presenting to the reader the pathogenicity of individual bacteria in Table 3 based on an extensive literature review is crucial for the relevance of the work.
Discussion
Question 7
Line no-247: Better to move these first 3 sentences into the introduction section.
Answer: We would like to thank you for the comment. The manuscript has been modified according to the Reviewer’s suggestions.
Question 8
Line no-255: Please delete these words "pathogens that are".
Answer: We would like to thank you for the comment. The fragment has been deleted.
Question 9
Line no-274: Please mention the name of the dental units in the sentence.
Answer: We would like to thank you for the comment; names of the dental units have been added to the manuscript.
Reference
Question 10
Line no-372, 442, and 491: Please re-check your references 3, 31, and 49 and write properly.
Answer: We would like to thank you for the comment, the references have been corrected.
Question 11
Line no-544: Year and DOI are missing. Please write it properly.
Answer: We would like to thank you for the comment. The references have been corrected.
